# The Effect of Satisfaction with Environmental Performance on Subjective Well-Being in China: GDP as a Moderating Factor

**Xinghua Zhao and Zongfeng Sun** * 

School of Political Science and Public Administration, Shandong University, Qingdao 266237, China; zhaoxinghua357@163.com

* Correspondence: sunzongfeng2017@sdu.edu.cn

**Abstract:** The purpose of this study was to examine the effect of environmental performance on subjective well-being against the background of different levels of economic development in China. The findings from the CGSS2015, combined with environmental quality data using the multi-level linear regression analysis method, indicated that the public's satisfaction with environmental performance will significantly enhance their happiness. The GDP variable was found to moderate this effect with reference to the expectation theory, positing that people have high expectations of happiness in provinces with a high GDP. The higher their expectations of being happy, the smaller the effect of satisfaction with environmental performance on happiness. These findings make contributions to both theory and public policy making, with relevant guidelines regarding physical activity recommendations and behavioral management strategies discussed.

**Keywords:** subjective well-being; satisfaction with environmental performance; multi-level linear regression analysis method

## 1. Introduction

While China's economic growth has seen great advances since the country's reform and opening up in 1978, the level of life satisfaction reported by the public has not simultaneously increased [1]. Prior studies show that income level [2–4], social support [5], and demographic variables such as age [6], marital status [7], employment status [8] and education level [9] significantly influence the degree of happiness that people feel. In recent research, the government has been seen to play an important role in making the public happy by delivering a good quality of public service [10], enhancing people's trust towards governments [11], and making the government accountable [12]. According to the research by Smith [2], it is the problem of environmental pollution that diminishes the happiness of urban residents in China. Although research by Whitely et al. indicates that the government's performance has had a positive impact on life satisfaction in Britain [13], the effect of environmental performance on subjective well-being in China has not been tested.

It is obvious that the high speed growth of China's economy is at the cost of environmental pollution, especially over-consumption of natural resources such as coal, petroleum, and natural gas. Therefore, a good personal and natural environment on which sustainable development depends apparently has not kept pace with the fast changing GDP. The environmental performance evaluated both by subjective and objective measures indicates the public's psychological process of cognition, judgement, and reflection on environmental pollution. Outcome of the psychological process will influence subjective well-being in a micro-level perspective. Therefore, this research innovatively combines the macro-level variable of environmental performance and the micro-level variable subjective

well-being, examining the effects of environmental perception on psychological process. As a typical country in East Asia, China is making great efforts to improve the quality of environment, which will generate significant impacts on the public's attitudes towards environment and happiness. Therefore, a culture on development, environmental protection, and happiness is gradually taking its shape, which can add to the discussion on the harmonization of natural and cultural resources across countries and culture. What's more, the comprehension on the relationship between environmental performance and subjective well-being in China will benefit the formulation, implementation, and evaluation of environment and energy policies.

In fact, various studies in the literature have reached the consistent conclusion that environmental pollution could reduce the level of residents' happiness, whether in developed or developing countries. Ferreira et al. analyzed the relationship between air quality and subjective well-being, finding negative impacts of $SO_2$ concentrations on self-reported life satisfaction [14] using a dataset combining the European Social Survey with a new one on environmental quality. In Spain, air pollution was found to be a significant factor explaining regional variations in happiness [9]. Examining real-time information on the location and well-being of over 20,000 participants in UK, another study found that participants were significantly happier in outdoor environments with green or natural habitats than in urban environments [15].

As Smyth [2] pointed out, air pollution could decrease the level of happiness, which was also confirmed by Zhang's research positing that air pollution will decrease the level of happiness and increase the probability of depression [16]. In addition, there is also a study that finds that $PM_{2.5}$ will significantly decrease the level of happiness, using a big data analysis method to analyze 0.21 billion microblog posts in China. In summary, it can be argued that environmental pollution will damage residents' happiness in China, in turn implying that a better environmental performance could benefit the public's happiness in urban and rural areas in China.

In order to deal with the serious problem of environmental pollution, the Chinese governments have invested much capital, human resources, and equipment into environmental governance. For instance, the central government reformed government agencies by re-organizing the Ministry of Ecology and Environment in 2008 in order to integrate the functions from various Ministries, such as the National Development and Reform Commission, Ministry of Natural Resources, Ministry of Agriculture, and so on. The money invested by the central government alone into environmental protection reached 63.3 billion yuan in the past five years (Source: http://www.xinhuanet.com/politics/2018lh/2018-03/17/c_137045643.htm), the details of which can be seen in Figure 1. According to the report released by the Ministry of Ecology and Environment in 2018, there are 121 cities taking up 35.8% of all the number of cities in China, whose air quality was up to the national standard. In addition, the pollution of surface water has been controlled by the evidence that the I~III water represented 71%, 3.1% higher than in 2017, while the proportion of V water was 6.7%, 1.6% lower than in 2017.

According to the New Public Management theory developed in the 1980s, citizens' satisfaction with public services may be highly correlated with the input, process and output of public affairs [17], such as public safety [18], public health [19], the public environment [20], public education [21], public traffic [22], and so forth. In addition, the New Institutional theory argues that the public authority's legitimacy largely depends on the public's recognition of its performance in areas such as economic growth, large projects, and social stability [23,24]. Therefore, it might be inferred that the governance performance in the realm of the public environment may also be underpinned by the theory of New Public Management and New Institutional theory. Here, we propose that the higher the environmental performance of a certain area, the higher the level of subjective well-being of its residents, on average.

Consistent with the Easterlin Paradox [25], a higher income is always associated with higher happiness scores. However, the relationship between GDP growth and subjective well-being over time seems to vary from negative to positive. In countries undergoing a transition, like China, the average life satisfaction scores do mirror the changes in GDP for at least the first ten years of the transition process [26]. According to the Environmental Kuznets Curve (EKC) theory, there exists

an Inverted-U relationship between pollution and economic development [27]. In the early stages of economic growth, degradation and pollution increase, but beyond a particular level of income per capita (which will vary for different indicators) the trend reverses, so that at high income levels economic growth leads to environmental improvement.

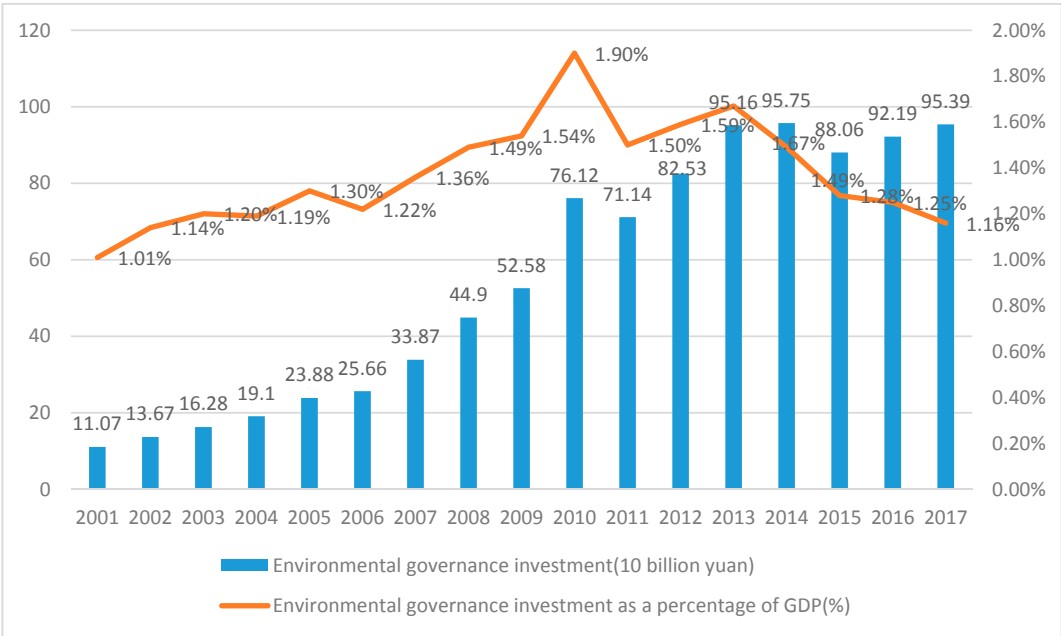

**Figure 1.** Investment in environmental governance and its share of GDP. Data Source: China Statistical Yearbook 2002–2018.

Therefore, in investigating the relationship between environmental performance and happiness, the effects generated by economic development need to be addressed. In this article, we treat the GDP variable as a moderator to mirror the various effects of environmental performance on subjective well-being. We won't take GDP as a mediating variable to examine the relationship between economic development and environmental protection. We argue that as GDP increases, the positive effects of environmental performance on subjective well-being will decrease, based on the notion that the public has a high expectation of happiness in those places where GDP is high [28,29], and low expectations when GDP is low. Normally, a high GDP brings with it a good quality of education and a public that is more critical towards public affairs. In other words, the high expectation pertaining to happiness will lead the public not to be easily satisfied with the current status, meaning that their standards of happiness will also be raised by their high expectations. As shown in Figure 2, environmental performance directly influences subjective well-being, and this effect will be moderated by the variable GDP.

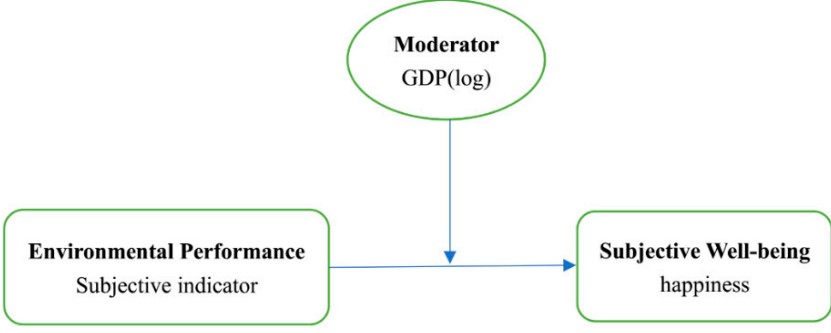

**Figure 2.** Research model.

## 2. Materials and Methods

### 2.1. Data Sources and Variables Specification

The dataset used in this article can be divided into two sections: individual level data and province level data. The former is a subset of the CGSS2015, which is a national survey conducted by Renmin University. The CGSS2015 covers 28 provinces using the stratified probability random sampling method, with a sample size of 10,968 (More details can be retrieved from its official website: http://www.cnsda.org/). Normally, the CGSS2015 includes numerous variables such as demographic variables, satisfaction with environmental protection, and self-reported happiness, among others. The data of CGSS2015 used in this article is nationally represented in 2015.

Moreover, the province level data are statistical data retrieved from the *Statistical Yearbook 2015* and the *Environmental Statistical Yearbook 2015*. Here, the data are all at provincial level, such as GDP, green park land per capita, garbage disposal, and wastewater volume.

- Subjective Well-being. In line with previous studies [30,31], we took happiness as the proxy variable for subjective well-being. In the CGSS2015 survey, the relevant question is: "Generally speaking, do you think your life is happy?' Possible responses include: "not at all happy", "a little happy" and "very happy", reported on a five-point Likert scale, whereby a score of "1" denotes "not at all happy", "3" "a little happy", and "5" "very happy".

- Environmental performance. As widely used in previous research [32], the subjective method was applied in this study to measure the efforts that governments had made to protect the environment in China. The CGSS2015 contains a question that could be used directly to measure environmental performance in subjective manner, namely: "Are you satisfied with the work that the government has done in protecting the environment?" Responses to this question were also ranked on a five-point Likert scale ranging from 1 to 5, with a higher number indicating greater satisfaction.

- Control variables. This study took account of all of the variables that might influence the variation of happiness as control ones, in order to derive the net effects of environmental performance on subjective well-being. Gender was treated as dummy variable [33], with female marked as "1" and male as "0". Age was set as a continuous variable, with age2 being the square term [34]. Years of education [35] was recoded according to participants' education experience; for instance, six years of education was recoded if the respondent reported their education background to have been primary school. Similarly, middle school was recoded as "9", high school as "12", undergraduate as "16", and postgraduate and above as "19", never attend school as "0". We took the log term of individual annual income [36] as a key control variable in light of previous research findings that income significantly influences subjective well-being. Ethnic group, faith [37], and place of residence were three dummy variables applied to control the variation of happiness among Han and minority groups, with and without religion, in urban and rural areas. The political party affiliation [37] variable was divided into four categories, namely, the Mass Group, the Youth League, the Democratic Party, and CPC member. Self-reported health conditions [38], measured on a five-point Likert table, was also controlled. Marriage status, such as being married, unmarried, divorced or widowed was also taken into consideration. Finally, social support [39] such as contact with neighbors and friends, and social trust were also controlled for. In addition, we also controlled for certain variables at the provincial level, such as garbage disposal, green park land per capita, wastewater, and environmental expenditure per capita [16].

Table 1 details each variable by name and shows its observation, mean, standard deviation, and minimum and maximum value.

**Table 1.** Descriptive variable analysis.

| Variables | Observation | Mean | Standard Deviation | min | max |
|---|---|---|---|---|---|
| Happiness | 10,953 | 3.867 | 0.821 | 1 | 5 |
| *Envi_sat* | 10,820 | 3.300 | 0.910 | 1 | 5 |
| Female | 10,968 | 0.531 | 0.499 | 0 | 1 |
| Age | 10,968 | 50.40 | 16.90 | 18 | 95 |
| Age2 | 10,968 | 2825 | 1742 | 324 | 9025 |
| Years of education | 10,939 | 8.690 | 4.710 | 0 | 19 |
| Income(log) | 8722 | 9.770 | 1.270 | 3.910 | 16.12 |
| Han | 10,948 | 0.922 | 0.268 | 0 | 1 |
| Religious | 10,822 | 0.109 | 0.312 | 0 | 1 |
| Rural areas | 10,968 | 0.410 | 0.492 | 0 | 1 |
| Party status | | | | | |
| Youth League member | 10,921 | 0.050 | 0.218 | 0 | 1 |
| Democratic party member | 10,921 | 0.001 | 0.038 | 0 | 1 |
| CPC member | 10,921 | 0.103 | 0.305 | 0 | 1 |
| Health | 10,961 | 3.610 | 1.070 | 1 | 5 |
| Marriage | | | | | |
| Married | 10,968 | 0.784 | 0.411 | 0 | 1 |
| Divorced | 10,968 | 0.020 | 0.143 | 0 | 1 |
| Widowed | 10,968 | 0.092 | 0.289 | 0 | 1 |
| Neighbor_contact | 9854 | 4.530 | 2.030 | 1 | 7 |
| Friend_contact | 9811 | 4.320 | 1.790 | 1 | 7 |
| Social trust | 10,927 | 3.470 | 0.960 | 1 | 5 |
| Garbage disposal(log) | 28 | 6.411 | 0.581 | 4.352 | 7.703 |
| Green park land per capita | 28 | 12.65 | 2.529 | 7.3 | 18.8 |
| Wastewater(log) | 28 | 12.39 | 0.650 | 10.04 | 13.72 |
| GDP(log) | 28 | 10.81 | 0.390 | 10.18 | 11.56 |
| *Envi_per* | 28 | 2.924 | 1.969 | 1.271 | 9.914 |

Note: *Envi_sat* and *Envi_per* represent satisfaction with environment performance and government environmental expenditure per capita, respectively.

*2.2. Methods*

The moderator variable, GDP, used in this article, was aggregated at province level, and some of our control variables such as garbage disposal, green area per capita, wastewater, and environmental expenditure per capita were applied as statistical variables at this level. In addition, it was necessary to examine the interaction effects between environmental performance and GDP, with a multi-level regression model seen as the most unbiased, efficient and consistent estimation method [40]. Our dependent variable adopted a five-point Likert table form, which determined the model selection in two ways. First, we considered the dependent variable to be a continuous one, meaning that our model would be the normal two-level linear regression model. The function of this model is shown below:

$$Y_{ij} = \beta_{0j} + \beta_{1j}X_{ij} + e_{ij}$$
$$\beta_{0j} = \gamma_{00} + \gamma_{01}W_j + \mu_{0j}$$

(1)

$Y_{ij}$ denotes the happiness score for an individual observation at Level 1 (subscript *i* refers to an individual case, subscript *j* refers to the province); $X_{ij}$ represents the Level 1 predictor; $\beta_{0j}$ refers to the intercept of the happiness variable in province *j* (Level 2); $\beta_{1j}$ denotes the slope for the relationship in province *j* (Level 2) between the Level 1 predictor and the dependent variable;$e_{ij}$ represents the random errors of prediction for the Level 1 equation (also sometimes referred to as $\gamma_{ij}$). $\gamma_{00}$ Signifies the overall intercept; this is the grand mean of the scores on the happiness variable across all provinces when all of the predictors are equal to 0. $W_j$ denotes the Level 2 predictor; $\gamma_{01}$ represents the overall regression coefficient, or the slope, between the happiness variable and the Level 2 predictor. $\mu_{0j}$ refers to the random error component in the deviation of the intercept of a group from the overall intercept.

Second, we also took the dependent variable as the ordinal variable, meaning that the model had to be fitted by multilevel mixed-effects ordered logistic models. For this purpose, we considered the two-level model where, for a series of *M* independent clusters and conditional on a set of fixed effects $X_{ij}$, a set of cut-points κ, and a set of random effects $\mu_j$, the cumulative probability of the response being in a category higher than *k* could be expressed as follows:

$$\Pr(y_{ij} > k | X_{ij}, \mathrm{k}, \mu_j) = H(X_{ij}\beta + Z_{ij}\mu_j - \mathrm{k}_k) \tag{2}$$

for j = 1, ... , M clusters, with cluster *j* consisting of *i* = 1, ... , $n_j$ observations. The cut-points k are labeled $\mathrm{k}_1, \mathrm{k}_2, \ldots, \mathrm{k}_{k-1}$, where K is the number of possible outcomes. *H* (·) is the logistic cumulative distribution function that represents cumulative probability.

## 3. Results

In order to show the variation of happiness among provinces, we constructed the map shown in Figure 3. The score of each province is calculated according to the weighted mean value. As can be seen from the figure, Beijing, Hebei, Inner Mongolia, Qinghai and Shandong emerged as the top five happiest provinces, on average. However, Xinjiang, Tibet, Taiwan, Hainan, Hangkong and Mocow were not able to be covered by the survey; therefore, missing values are shown. The people living in Guangxi, Sichuan and Hubei emerged as less happy than those in Inner Mongolia, Shandong, Hebei and Qinghai.

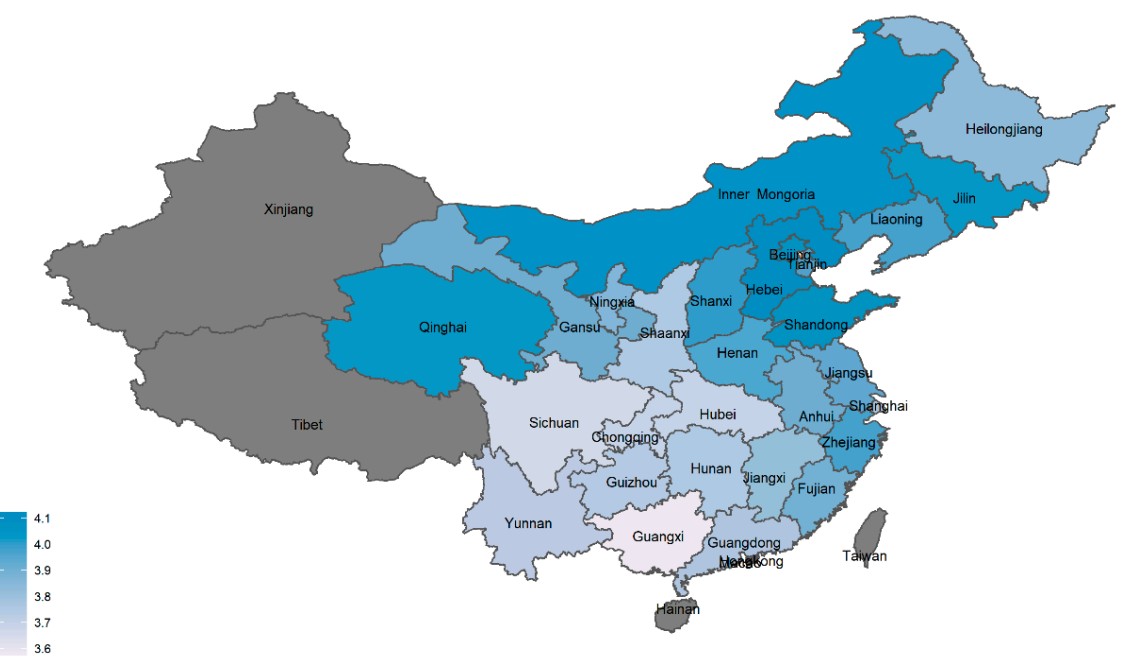

**Figure 3.** Distribution of Happiness. Note: Gray areas represent the missing values; the bluer, the happier. Source: Map generated on the data from CGSS2015.

Alongside this, Figure 4 demonstrates the variation of satisfaction with environmental performance among the provinces. The study found that citizens in Inner Mongolia, Shan' xi, Qinghai and Gansu province were more satisfied with environmental performance, whereas those in Guangdong, Liaoning and Beijing reported a low level of satisfaction with this. According to both Figures 3 and 4, there seems to be a positive link between satisfaction with environmental performance and happiness in China. In order to discern the net effect of satisfaction with environmental performance on happiness under the condition of varying levels of economic development, we constructed a multi-level linear regression model, as seen in Table 2.

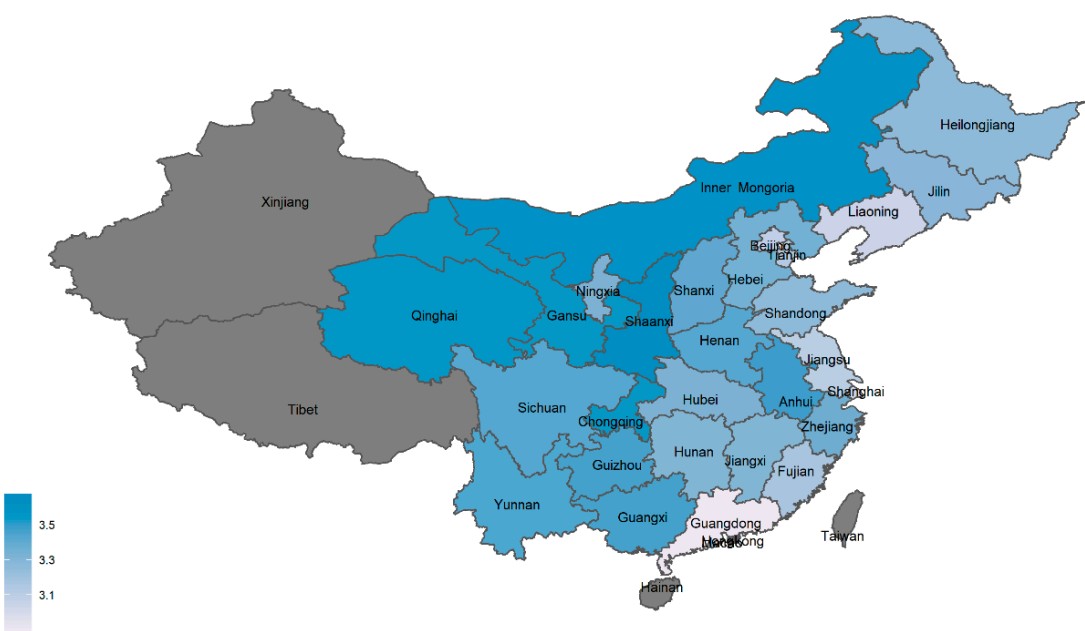

**Figure 4.** Distribution of Environmental Satisfaction. Note: Gray areas represent the missing values; the darker the shade of blue, the more satisfied the participants reported feeling. Source: Map generated on the data from CGSS2015.

**Table 2.** Two-level Linear Regression Model (random intercept model).

| | | | Robust Standard Error | |
|---|---|---|---|---|
| **Happiness** | **(1)** Ordinal | **(2)** Continuous | **(3)** Ordinal | **(4)** Continuous |
| *Envi_sat* | | | | |
| Dissatisfied | 6.082 *** | 1.905 ** | 6.078 *** | 1.905 ** |
| | (2.76) | (2.55) | (2.76) | (2.55) |
| Somewhat satisfied | 4.392 ** | 1.442 ** | 4.385 ** | 1.442 ** |
| | (2.08) | (2.03) | (2.08) | (2.03) |
| Satisfied | 5.887 *** | 1.921 *** | 5.883 *** | 1.921 *** |
| | (2.82) | (2.74) | (2.82) | (2.74) |
| Very satisfied | 5.020 * | 1.236 | 5.019 * | 1.236 |
| | (1.92) | (1.41) | (1.92) | (1.41) |
| GDP(log) | 1.141 *** | 0.394 *** | 1.141 *** | 0.394 *** |
| | (2.89) | (3.18) | (2.89) | (3.18) |
| Interaction | | | | |
| Dissatisfied*GDP(log) | −0.577 *** | −0.176 ** | −0.577 *** | −0.176 ** |
| | (−2.67) | (−2.39) | (−2.67) | (−2.39) |
| Somewhat satisfied*GDP(log) | −0.406 ** | −0.128 * | −0.405 * | −0.128 * |
| | (−1.96) | (−1.83) | (−1.96) | (−1.83) |
| Satisfied*GDP(log) | −0.522 ** | −0.165 ** | −0.522 ** | −0.165 ** |
| | (−2.54) | (−2.39) | (−2.54) | (−2.39) |
| Very satisfied*GDP(log) | −0.396 | −0.0885 | −0.395 | −0.0885 |
| | (−1.52) | (−1.01) | (−1.52) | (−1.01) |
| Control variables | | | | |
| *Envi_per* | 0.0218 | 0.00621 | 0.0230 | 0.00621 |
| | (0.42) | (0.40) | (0.44) | (0.40) |
| Wastewater(log) | −1.024 ** | −0.334 ** | −1.026 ** | −0.334 ** |
| | (−2.30) | (−2.46) | (−2.30) | (−2.46) |
| Garbage disposal(log) | 0.174 | 0.0471 | 0.175 | 0.0471 |
| | (0.63) | (0.56) | (0.63) | (0.56) |
| Green park land per capita | 0.0411 * | 0.0112 | 0.0408 * | 0.0112 |
| | (1.66) | (1.48) | (1.65) | (1.48) |

**Table 2.** *Cont.*

| Happiness | (1) Ordinal | (2) Continuous | (3) Ordinal | (4) Continuous |
|---|---|---|---|---|
| | | | **Robust Standard Error** | |
| Female | 0.313 *** | 0.0996 *** | 0.313 *** | 0.0996 *** |
| | (6.29) | (5.55) | (6.30) | (5.55) |
| Age | −0.0628 *** | −0.0239 *** | −0.0628 *** | −0.0239* ** |
| | (−6.23) | (−6.59) | (−6.24) | (−6.59) |
| Age2 | 0.000788 *** | 0.000292 *** | 0.000788 *** | 0.000292 *** |
| | (8.30) | (8.59) | (8.30) | (8.59) |
| Years of education | 0.0279 *** | 0.0104 *** | 0.0279 *** | 0.0104 *** |
| | (3.81) | (3.95) | (3.81) | (3.95) |
| Income(log) | 0.137 *** | 0.0544 *** | 0.137 *** | 0.0544 *** |
| | (5.43) | (5.99) | (5.42) | (5.99) |
| Han Group | −0.0410 | −0.0207 | −0.0430 | −0.0207 |
| | (−0.39) | (−0.55) | (−0.41) | (−0.55) |
| Religion | 0.420 *** | 0.151 *** | 0.423 *** | 0.151 *** |
| | (5.08) | (5.10) | (4.89) | (5.10) |
| Rural areas | 0.0794 | 0.0299 | 0.0793 | 0.0299 |
| | (1.32) | (1.38) | (1.32) | (1.38) |
| Party status | | | | |
| Youth League member | 0.364 ** | 0.124 ** | 0.364 ** | 0.124 ** |
| | (2.55) | (2.46) | (2.55) | (2.46) |
| Democratic party member | 0.0257 | −0.0141 | 0.0244 | −0.0141 |
| | (0.05) | (−0.07) | (0.04) | (−0.07) |
| CPC member | 0.293 *** | 0.102 *** | 0.294 *** | 0.102 *** |
| | (3.75) | (3.59) | (3.75) | (3.59) |
| Marriage status | | | | |
| Married | 0.669 *** | 0.259 *** | 0.669 *** | 0.259 *** |
| | (6.53) | (6.94) | (6.53) | (6.94) |
| Divorced | −0.219 | −0.0877 | −0.218 | −0.0877 |
| | (−1.19) | (−1.30) | (−1.19) | (−1.30) |
| Widowed | 0.221 | 0.0735 | 0.220 | 0.0735 |
| | (1.61) | (1.48) | (1.60) | (1.48) |
| Health | 0.486 *** | 0.177 *** | 0.486 *** | 0.177 *** |
| | (18.26) | (19.22) | (18.26) | (19.22) |
| Social trust | 0.324 *** | 0.113 *** | 0.324 *** | 0.113 *** |
| | (12.30) | (12.24) | (12.30) | (12.24) |
| Neighbor_contact | 0.0136 | 0.00409 | 0.0136 | 0.00409 |
| | (0.96) | (0.80) | (0.96) | (0.80) |
| Friend_contact | 0.0696 *** | 0.0242 *** | 0.0695 *** | 0.0242 *** |
| | (4.45) | (4.33) | (4.44) | (4.33) |
| Constant | | 1.255 | | 1.255 |
| | | (1.42) | | (1.42) |
| Cut1 | 1.947 | | 1.976 | |
| | (0.70) | | (0.71) | |
| Cut2 | 3.890 | | 3.919 | |
| | (1.40) | | (1.41) | |
| Cut3 | 5.362 * | | 5.392 * | |
| | (1.92) | | (1.93) | |
| Cut4 | 8.636 *** | | 8.666 *** | |
| | (3.10) | | (3.11) | |
| Var(_cons[province])_cons | 0.0782 *** | | 0.0776 *** | |
| | (2.94) | | (2.93) | |
| N | 7551 | 7551 | 7551 | 7551 |

Note: t statistics are shown in parentheses; * *p* < 0.1, ** *p* < 0.05, *** *p* <0.01, Standard errors are clustered at province level.

As shown in Table 2, two approaches were used to estimate the effects of satisfaction with environmental performance on the public's subjective well-being, considering the dependent variable as ordinal and then continuous, separately. In addition, we overcame the problem of heterogeneity using the robust standard error shown in models 3 and 4. Some of the variables such as income, marriage status and health condition can be sensitive, thus the missing values show up in the model.

Keeping other aspects constant, all of the models in Table 2 show that GDP has a significantly positive impact on happiness, at a significance level of 0.01. In other words, as GDP increases by one unit in the log term, happiness in China will increase by 0.394 on the scale of 1–5, on average. Moreover, a positive and significant relationship emerged between satisfaction with environmental performance and happiness, with the significance level ranging from 0.05 to 0.01. Compared to the control group, those who are in the Dissatisfied group are 1.905 higher in happiness on average, at a significance level of 0.05. This finding was further confirmed by the estimation of the multilevel mixed-effects ordered logistic model 1 and model 3. Compared to the control group, those who reported being somewhat satisfied were found to score 1.442 higher in terms of happiness, at a significance level of 0.05, which was also confirmed by models 1 and 3 using different estimation methods. In addition, compared with the control group, those who reported feeling satisfied scored 1.921 higher in terms of happiness on average, at a significance level of 0.01. However, compared to the control group, those who reported being very satisfied showed no significant difference in terms of happiness, as found from models 2 and 4. Even though the results emerged as significant in models 1 and 3, the level of significance was quite low, at only 0.1.

When considering the moderating variable, GDP, the effect of satisfaction with environmental performance on happiness varied. As shown in Table 2, all four models show that compared to the control group, namely, the group of not at all satisfied * GDP, the coefficient of Dissatisfied * GDP is significantly lower, so the same with somewhat satisfied * GDP, and satisfied * GDP, except the very satisfied * GDP. In other words, as GDP increases, the effect of satisfaction with environmental performance on happiness steadily decreases.

In addition, wastewater emission did have a significant negative effect on happiness, whereas green park land areas can enhance happiness. Moreover, gender, age, years of education, whether participants belong to a certain religion, their political party affiliation, self-reported health condition and income have a significant impact on happiness, as proven in prior research. The social support factors such as marriage status, social trust, and contact with friends also emerged as having a significant influence on happiness.

Another aspect we needed to pay attention to was the causal relationship between environmental performance and subjective well-being. For instance, some will argue that it is because of the low level of subjective well-being that leads to low level of satisfaction with environment performance, which is reverse causality. It is acknowledged that no rigid causal relationship could be established without experimental study. We attempted to engage with this aspect by adding enough control variables, which have been examined before, and eliminating the problem of omitted variables. Second, we used a robust standard estimation method to control the problem of heterogeneity. Third, the control variables such as wastewater volume, green park land and environmental expenditure were all drawn from 2014 data, collected one year before our survey data, which could partly deal with the causal relationship problem.

Furthermore, in order to obtain a robust estimation, we conducted a weighted multi-level regression analysis [41]. We found that there was no significant difference between the weighted and unweighted multi-level linear regression methods in parameter estimation. In other words, the data used in this article was nationally representative without significant sampling error. Our findings are robust after considering the problem of sampling error.

## 4. Discussion

This research examines the effects of satisfaction with environmental performance on subjective well-being against the background of different levels of economic development in China. Using subset data from the CGSS2015, we found that the public's satisfaction with environmental performance enhanced their happiness, but that this effect varied in line with GDP. This is consistent with prior theories, in particular the New Public Management Theory and New Institutional Theory. This paper provides new evidence from China that the public are growing increasingly concerned about the quality of the environment and the improvement of environmental performance by the country's governments. When the public are satisfied with environmental protection, they will demonstrate more life happiness.

In addition, the interaction effects show that as the level of economic development is enhanced, the effects of satisfaction on happiness steadily decrease. The reason for this is that those who live in provinces with a high GDP usually have high expectations of happiness. In other words, they define happiness in a stricter way than those who live in provinces with a low GDP. In fact, the public services delivered by local governments are more attractive in high-GDP provinces, especially the public environment service, which highly depends on the public investment made by governments. A high level of happiness may thus be expected when local governments invest more funding into public environmental governance. In addition to this, these governments need to discern ways and methods of enhancing the public's satisfaction with public environmental governance.

Methodologically, although two different-level data are used in this study, we have to acknowledge that this isn't an innovation in research method because there are earlier studies on subjective well-being using linked survey and register data [42]. We have to emphasize its advantage of this research method of hierarchical linear model. As mentioned above, the research method used in this article can make variables from macro-level (objective measure for environment performance) and micro-level data interact under different conditions, namely examining the relationship between environmental performance and subjective well-being under various conditions of GDP.

Research findings made in this paper can apply to those developing countries where the GDP per capita and environment pollution are similar to China's in 2015. In addition, the results presented in this article can also be meaningful to those countries who try to enhance the citizen's subjective well-being by governing the environmental pollution. Last but not the least, the findings on relationship between environmental performance and subjective well-being under various conditions of GDP has implications for countries with different cultures from Asia. Countries in the world who attempt to enhance their citizen's subjective well-being have to consider the effects of GDP growth and environmental protection.

## 5. Conclusions

In China, the public is growing increasingly concerned about environmental quality, clean air, green water, and the beauty of the areas in which they live, which can determine their quality of life, that is, their subjective well-being. Although the Chinese government has made great efforts to improve the quality of the environment by investing substantial amounts of money into this area, the effect of environmental performance on happiness is yet to be known. This paper finds that the public's satisfaction with environmental performance will significantly enhance their level of happiness. The study also shows that the GDP variable moderates this effect through the assumptions of the expectation theory of happiness, namely, that people will have high expectations in high-GDP provinces. The higher the expectation pertaining to happiness, the smaller the effect of satisfaction with environmental performance will be. These findings make a new contribution by testing the effect of environmental performance on happiness in the context of China, and examining the validity of the New Public Management and New Institutional theories. In practice, more public policies need to be formulated in order to improve the quality of the environment, particularly in dealing with wastewater and air pollution. Policies aiming at enhancing the public's subjective well-being should be diversified

considering their happiness expectation caused by GDP. Subjective well-being shouldn't only rely on GDP growth, and attention paid to environment protection also matters. Moreover, the combining effects of environmental protection and GDP growth on subjective well-being should draw policy makers' attention.

This paper also has limitations. First, climate change factor can't be controlled in this article because of the air quality data missing at province level, which can be studied in the future. Second, the link of environmental protection and subjective well-being isn't rigid causality relationship. Future researches can fill the gap by using experiment research method or finding instrumental variable to deal with this problem.

**Author Contributions:** Conceptualization, X.Z. and Z.S.; methodology, X.Z.; validation, Z.S.; formal analysis, X.Z.; investigation, X.Z.; resources, X.Z.; data curation, X.Z.; writing—original draft preparation, Z.S.; writing—review and editing, Z.S.; visualization, X.Z.; supervision, X.Z.; project administration, Z.S.; funding acquisition, Z.S. All authors have read and agreed to the published version of the manuscript.

**Funding:** This research received the National Social Science Fund of China (Grant Number: 18CGL037).

**Conflicts of Interest:** The authors declare no conflict of interest.

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
