# Peer review of "The Effect of Satisfaction with Environmental Performance on Subjective Well-Being in China: GDP as a Moderating Factor"

_sustainability, doi:10.3390/su12051745_

Round 1

Reviewer 1 Report

Comments

Figure 1 could include a longer time span in order to get a broader perspective on the issue. Are the data nationally representative or not? This issue should be stated explicitly in the data section. Does the data have a panel structure that could be used in the estimation of the links? The paper should acknowledge that there are earlier studies on subjective well-being using linked survey and register data (https://doi.org/10.1177/001979391206500203). Are standard errors clustered or not? What is the external validity of the results that are presented in the paper? The paper does not consider the heterogeneity in links. The links that are estimated can differ significantly e.g. by age/gender. The paper should discuss more practical policy lessons.

Reviewer 2 Report

Overall a well written and interesting paper, in my view of interest to readers. Implications for policy are included, but somewhat briefly. Below I suggest some places where this might be strengthened, together with some other issues. 

Pages 52-63 and Fig 1: data presented indicate that the level of investment in environmental protection is not keeping pace with growing GNP. This is especially significant when nearly 2 in 3 cities suffer from poor air quality. Does this explain in part why as GNP grows higher, satisfaction with environmental performance falls?

Line 71 Insert 'might' be inferred, delete 'can be inferred'.

Lines 84-85 How does EKC fit in with the findings? Was there evidence to support EKC theory? Not mentioned in discussion or conclusions [see next point].

Lines 86-95 Objectives should be stated more clearly: this might give you a better guide to matching up the content of later sections.

Fig 3: The fullness and accuracy of official data are not evaluated. Might this differ by region?

Fig 3: You stated earlier that people living in cities on average were less happy than those living in rural areas. Might this influence the spatial pattern evident in the figure?

Table 1: Climate change is not included as a variable: no data?

Line 212 Insert 'the' after 'who are in'.

Line 234 Insert 'is acknowledged' and delete 'should be acknowledged'.

Lines 240-141 Brief explanation needed for this point.

Lines 242-244 Brief explanation needed regarding the implications of this statistical finding.

Line 276 Very generalised point, not clearly based in your data analyses, perhaps related to Fig 1?

Conclusions: Are there weaknesses you see in your data and analysis methods which you could point out, as a guide to others who might want to replicate your methods?
